# Critical Video-Language Understanding via Query-Guided Frame Selection and Visual-Query Transformation

## Abstract

Recent advances in language-model-based video understanding have developed rapidly, driven by the emergence of LLMs to a great extent. Nevertheless, existing research has primarily concentrated on creating a projection mechanism converting video features into tokens—a method that is both conceptually simplistic and practically ill-performed. In this study, we propose `VaQuitA`, a brand new framework which more effectively unifies video representations and text inputs. At the data level, instead of sampling frames uniformly, we take advantage of a CLIP (Radford et al., 2021)-score-based scheme, ensuring frames are more closely aligned with the query. At the feature level, we introduce a tunable Video Perceiver and a Visual-Query Transformer (VQ-Former), which together improve the synergy between the input question and the video features. In addition, we show that adding a brief prompt—specifically, "Please be critical." improves the LLM's ability to comprehend video content. Experiments across various benchmark datasets show that `VaQuitA` establishes a new state-of-the-art for zero-shot video question-answering, while also enabling high-quality multi-turn video-based dialogues with users. Code will be released.

## 1 Introduction

The rise of deep learning tools for video interpretation has ushered in significant progress in video-centric tasks (Xu et al., 2021; Wang et al., 2022b; 2023). Yet, current models for video comprehension often falter when engaging in spontaneous discussions about video content (Zhong et al., 2022). A dialogue system rooted in video content can transform video searches, enhance monitoring techniques, and assist in summarizing pivotal events. Importantly, it offers a unified, accessible interface for video tasks, including action recognition, location identification, detection and tracking (Mu et al., 2023). This proficiency is especially noteworthy, highlighting the model's ability to understand temporal and spatial indications, grasp context, and perceive extended relationships (Liu et al., 2023d).

Existing research in Large Video Language Models (Yang et al., 2022; Zhang et al., 2023; Gao et al., 2023; Li et al., 2023b; Maaz et al., 2023; Liu et al., 2023c) predominantly adopts a uniform sampling strategy for frame selection. These models typically use a single projection layer to transfer and align video semantic content into the token space. The resulting tokenized video embeddings are then concatenated with query embeddings and fed into LLMs for response generation. However, this straightforward approach fails to adequately guide the projection of video features into specific text representations or sufficiently highlight which spatial or temporal aspects of the video should be emphasized. Given the constraints of limited training data, this methodology often leads to suboptimal performance in out-of-distribution video understanding tests (Maaz et al., 2023). In real-world scenarios, this can lead to perplexing errors in video conversation systems (Liu et al., 2023c).

To mitigate the above problems, we introduce `VaQuitA`, an innovative framework that redefines the approach to video and textual information integration. `VaQuitA` diverges from traditional methodologies by implementing a CLIP (Radford et al., 2021)-score guided frame sampling method. This innovation allows for the selection of frames that exhibit a higher relevance to the input question, thereby addressing the limitations of

uniform frame sampling. The framework further advances the interaction between video content and textual queries through the integration of a trainable Video Perceiver. This component enhances the processing of video features, ensuring a more nuanced understanding of the visual content. Complementing this is our Visual-Query Transformer (VQ-Former), which acts as a pivotal element in aligning the video features with the textual query, facilitating a more coherent and context-aware interplay. Furthermore, `VaQuitA` incorporates a novel approach in its interaction with LLMs. By introducing a simple, yet effective prompt — "Please be critical." — into the LLM input during testing, we notice a marked enhancement in the model's capability to interpret video material. This refinement leads to a more critical and discerning analysis by the LLM, enhancing its performance in complex video understanding tasks. The contributions of the paper are:

- We propose `VaQuitA`, a novel video understanding model that strengthens the alignment of text features and video features. The alignment lies in both the raw data level and the feature level, which enhances the fusion of question and video information, leading to stronger reasoning ability of the video question answering model.

- We uncover the fact that adding an additional prompt, "Please be critical.", before the question can improve the understanding ability of `VaQuitA`.

- Our proposed `VaQuitA` achieved SOTA performance on the Zero-shot Video Question Answering task. It can also conduct top-notch multi-turn conversations.

## 2 Related Works

We briefly summarize existing works in the related areas of video conversation, vision large lanugage models, and visual-text alignment.

**Video Conversation.** With the rapid development of LLMs, researchers begin to transfer their extraordinary reasoning abilities to the video conversation area (Song et al., 2023; Kim et al., 2024; Maaz et al., 2024; Liu et al., 2024; Li et al., 2023d; Wu, 2024; Xu et al., 2024; Fei et al., 2024; Hong et al., 2024). The SeViLA framework (Yu et al., 2023) leverages BLIP-2 (Li et al., 2023a) for both temporal keyframe localization and question answering in videos. VideoChat (Li et al., 2023b) integrates foundational video models and LLM using a learnable neural interface. Video-ChatGPT (Maaz et al., 2023) develops a multimodal model that merges a video-adapted visual encoder with a large language model. Despite the progress, the current video conversation capability is still limited due to the insufficient exploitation of question and video interplay.

**Vision Large Language Models.** Recent progress in computer vision has been propelled by the emergence of groundbreaking vision-language models. These models mark a considerable step forward in developing versatile vision models that can handle multiple tasks at once (Gupta et al., 2022; Maaz et al., 2022). A standout model in this realm is CLIP (Radford et al., 2021), trained on 400 million image-text pairs, showcasing exceptional zero-shot capabilities across many benchmarks. In more recent times, Flamingo (Alayrac et al., 2022) is a new family of Visual Language Models designed to rapidly adjust to novel tasks using a minimal number of annotated examples. It proposes perceiver resampler and gated cross-attention architectures. BLIP-2 (Li et al., 2023a) represents an effective method for pre-training that leverages existing image encoders and language models that have undergone pre-training. While some of these models are compatible with both images and videos, videos that are minute-level are yet challenging to process with input questions for accurate answers, and there is a growing demand for a robust large video-language model.

**Visual-Text Alignment.** Contemporary progress in aligning visual-text features primarily revolves around the concept of harmonizing multimodal features originating from various representational spaces. The foundational work (Duan et al., 2022) highlighted the challenges of aligning evolving features during training. Progressing from this, the Multi-Modality Cross Attention Network (Wei et al., 2020) and the "MVPTR" framework (Li et al., 2022b) both emphasized the significance of fine-grained feature alignment and cross-modal interactions. Further innovations in multimodal fusion were proposed through Central-Net (Vielzeuf et al., 2018) presenting a multilayered integration approach, and ADAPT (Lin et al., 2022) which introduced dynamic action-based context alignment for Vision-Language Navigation. Contrasting

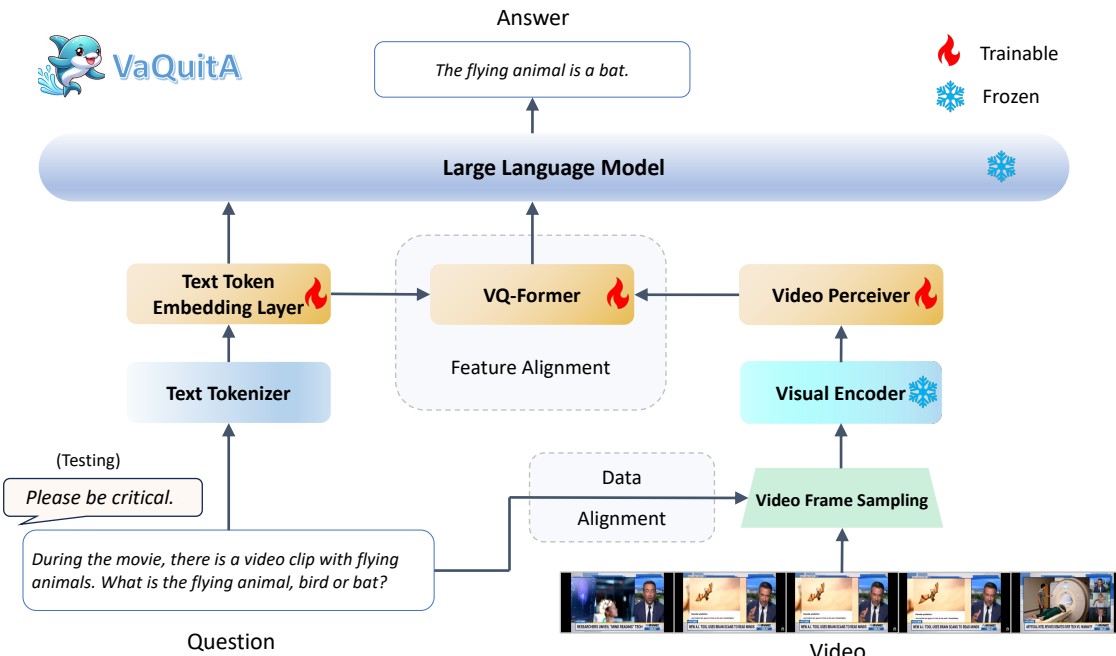

Figure 1: **Framework overview.** In response to a specific question, our framework begins by processing the input video with a sampling module that identifies key frames based on their relevance to the question's context. These frames are then processed by a pre-trained visual encoder to obtain spatio-temporal features. These features are subsequently refined into condensed embeddings by our newly developed Video Perceiver. In parallel, the question undergoes tokenization. Both the video and text embeddings are then synergized using our Visual-Query Transformer, which aligns the multimodal information more effectively. The resulting text-influenced video features are concatenated with the text embeddings and fed into the Large Language Model to generate the answer. During the testing phase, we propose to add an additional prompt, "Please be critical.", before the question for performance enhancement. The whole framework supports end-to-end training.

with existing approaches, our method enforces the alignment of video and text embeddings through a novel video feature resampling network and a bespoke cross-attention module tailored for the LLM input space. The resampling module enhances the alignment in the input raw data level and the cross-attention module strengthens the alignment in the feature learning level. Our approach represents an innovative direction in the field of large vision language modeling.

## 3   VaQuitA Framework

Our proposed `VaQuitA` framework consists of three novel components: Data Alignment module (Sec. 3.1), Feature Alignment module (Sec. 3.2), and test-time Prompt Engineering (Sec. 3.4). The entire pipeline is illustrated in Fig. 1.

### 3.1   Data Alignment

Existing methodologies typically employ a uniform sampling approach to extract frames for video conversation (Maaz et al., 2023; Bhattacharya et al., 2023; Yang et al., 2023a) or video understanding in general (Lin et al., 2019; Li et al., 2022a; Wang et al., 2022a). Such uniform sampling method, while straightforward, often results in the loss of critical information contained in the frames that are not selected, affecting the model's ability to understand videos effectively. To address this limitation, we present a new method in our `VaQuitA` that leverages the semantic similarity between the video frames and the question prompt for frame

selection. This technique ensures a more congruent alignment between the features of the question and those of the frames at the raw data level. We refer to this as the "Data Alignment" module.

**CLIP Feature Similarity-based Frame Selection for Training.** Given the input video of $L$ frames in total, instead of getting a certain number of frames with only uniform sampling, we also select frames based on the similarity between the frame features and the input query. Suppose we sample $T$ frames in total, we propose to select $\frac{T}{2}$ frames uniformly over the temporal dimension and another $\frac{T}{2}$ frames using the similarity-based approach. Specifically, we extract the text feature of the query using CLIP model, denoted as $f_{\text{query}}$, and the visual features of the remaining frames that are not selected as $\{f_{\text{video}}^1, f_{\text{video}}^2, \cdots, f_{\text{video}}^{L-\frac{T}{2}}\}$, the similarity is calculated as

$$\text{Cosine-Similarity}(f_{\text{query}}, f_{\text{video}}^i) = \frac{f_{\text{query}} \cdot f_{\text{video}}^i}{\|f_{\text{query}}\|_2 \times \|f_{\text{video}}^i\|_2}, \tag{1}$$

and we select the indices of the top $\frac{T}{2}$ values. The motivation is that uniform sampling will lead to information loss due to its non-adaptivity, and by employing the proposed similarity-based approach, frames that are most related to the question will be selected, improving representation learning ability. Our proposed sampling strategy is different from some recent works, *e.g.*, using motion importance (Zhi et al., 2021), and dynamic sampling (Zheng et al., 2020), which are based on the inherent properties of the video frame statics. Our sampling approach makes the first attempt to employ the prompt-frame similarity as guidance for frame sampling, bridging the two modalities for effective modeling.

**Uniform Sampling for Testing.** Our proposed sampling method is implemented during the training phase of our model. For the testing or inference stage, we revert to uniform sampling **due to efficiency considerations and the need for speed in real-world applications. This approach is enough to demonstrate satisfactory performance in our experimental evaluations**. We supplement an example in the supplementary showing that our sampling approach can improve testing performance as well compared with uniform sampling, despite being slower.

## 3.2 Feature Alignment

Visual data are regarded as the reflection and capture of the physical world while text data can be seen as the abstract of the understanding of the world and the fundamental logic (LeCun, 2022). The successful alignment of visual and textual information is significant for an intelligent system to work appropriately. Instead of directly concatenating the tokenized text and visual features to put into LLM (Liu et al., 2023b; Maaz et al., 2023; Zhang et al., 2023; Chen et al., 2023; Liu et al., 2023c), we propose a novel Visual-Query Transformer, abbreviated as VQ-Former, to produce text-guided video embeddings before concatenation with the text embeddings. The inspiration comes from recent work on visual-text pretraining (Li et al., 2023a; Alayrac et al., 2022), and the illustration of the architecture is provided in Fig. 2. Notice that although the self-attention mechanism in LLM already interacts text tokens with visual tokens to some extent by treating visual tokens as language tokens, our proposed feature alignment module treat text and video features as different domain features through cross-attention operations. This practice enhance the input visual token qualities by computing attention and attending over input query tokens.

### 3.2.1 Video Perceiver

Given a sampled video snapshot, we first apply a pretrained CLIP model to extract semantic features for each frame. Suppose the extracted spatio-temporal feature embeddings are $F \in \mathbb{R}^{T \times n \times d}$, where $T$ is the sampled frame number, $n$ is the number of features for each frame, *i.e.*, the patch number for CLIP model, and $d$ denotes the dimension of feature. To facilitate the alignment with text embeddings and input into the LLM, we need to resample and reduce the number of video features for computation feasibility. Inspired by Perceiver Resampler (Alayrac et al., 2022), we put forward Video Perceiver which transforms the spatio-temporal visual attributes into a number of learned output tokens. The spatio-temporal features are first added by Time Encodings of shape $T \times 1 \times d$ to store the sequence order information and then flattened to shape $Tn \times d$ for the cross-attention module dimension match. This cross-attention module

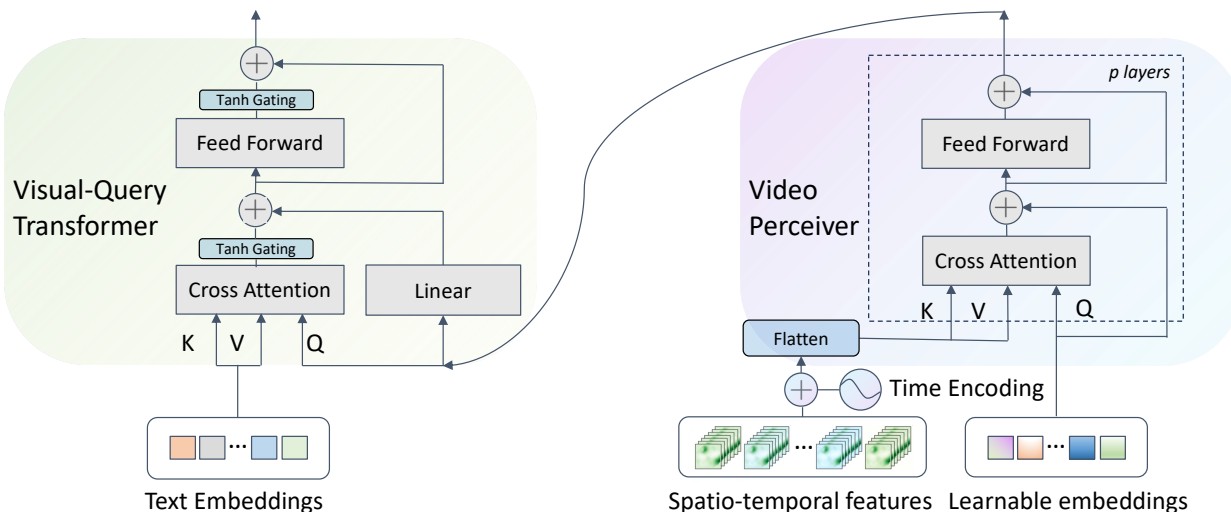

Figure 2: **Feature alignment**. The extracted spatio-temporal features of the video clip first go through Video Perceiver for representative embedding extraction, and are afterwards sent to Visual-Query Transformer for interleaving with text embeddings.

employs a collection of learned latent vectors to query (Q), while the keys (K) and values (V) combine the flattened spatio-temporal visual attributes with these learned latent vectors. The shape of the learned latent embeddings is $m \times d$, where $m$ denotes the number of latent embeddings. The weights of the learned latent embeddings are randomly initialized. Following transformer (Vaswani et al., 2017), feed-forward networks and residual connections are added for efficient modeling. The output embedding shape remains the same as the input learnable embeddings, *i.e.*, $m \times d$. We use $p$ to denote the number of layers of the Video Perceiver.

### 3.2.2 Visual-Query Transformer

The input question goes through a text tokenizer and a text token embedding layer and turns into query embeddings, which, together with the learnable embeddings output of the video perceiver, are sent into VQ-Former. The layers derive their queries from vision features, whereas the keys and values originate from the language inputs. Visual-Query Cross-Attention layer is applied for the query feature (denoted as $X \in \mathbb{R}^{l \times d_{text}}$) and video feature (denoted as $M \in \mathbb{R}^{m \times d}$) interleaving, where $l$ is the length and $d_{text}$ is the text embedding dimension.

**Visual-Query Cross Attention.** In the Visual-Query Cross Attention layer, we adopt a multi-head mechanism as in Transformer (Vaswani et al., 2017). We denote the head index as $h$ and the inner feature dimension of each head as $d_h$. Given the input learned video feature, we first apply Layer Normalization (Ba et al., 2016) and denote the normalized one as $M$. Note that although $M$ is learned, the existence of the Time Encodings guarantees that the temporal information of input frames is kept. Then we have the $Q, K, V$s for each head calculated as

$$Q^{(h)} = MW_Q^{(h)}/s_q, K^{(h)} = XW_K^{(h)}, V^{(h)} = XW_V^{(h)}, \tag{2}$$

and

$$O_a^{(h)} = \texttt{Softmax}(Q^{(h)}K^{(h)^\top})V^{(h)}W_O^{(h)}, \tag{3}$$

where $W_Q^{(h)} \in \mathbb{R}^{d \times d_h}$, $W_K^{(h)} \in \mathbb{R}^{d_{text} \times d_h}$, $W_V^{(h)} \in \mathbb{R}^{d_{text} \times d_h}$ and $W_O^{(h)} \in \mathbb{R}^{d_h \times d_{text}}$ are learnable weight parameters of head $h$. $s_q$ is a scaler representing the scale parameter, and $h$ represents the head index. Denoting the Visual-Query Cross Attention layer output as $O_a$, we have that $O_a$ is the multi-head concatenation of each head output:

$$O_a = \text{Concat}(O_a^{(1)}, \cdots, O_a^{(H)}), \tag{4}$$

where $H$ denotes the head number. The dot product attention computation aligns the semantics of the video embedding $M$ and query embedding $X$, contributing to the selection and learning of the visual features more

| Model | LLM Backbone | MSVD-QA | | MSRVTT-QA | | Activity Net-QA | | NExT-QA | |
|---|---|---|---|---|---|---|---|---|---|
| | | Acc. (↑) | Score (↑) | Acc.(↑) | Score(↑) | Acc.(↑) | Score (↑) | Acc.(↑) | Score (↑) |
| FrozenBiLM* (Yang et al., 2022) | DeBERTa-900M | 32.2 | – | 16.8 | – | 24.7 | – | – | – |
| VideoLLaMA† (Zhang et al., 2023) | Vicuna-7B | 51.6 | 2.5 | 29.6 | 1.8 | 12.4 | 1.1 | – | – |
| LLaMA-Adapter† (Gao et al., 2023) | LLaMA-7B | 54.9 | 3.1 | 43.8 | 2.7 | 34.2 | 2.7 | – | – |
| Video Chat* (Li et al., 2023b) | Vicuna-7B | 56.3 | 2.8 | 45.0 | 2.5 | 26.5 | 2.2 | 56.2 | 3.2 |
| Video-ChatGPT* (Maaz et al., 2023) | Vicuna-7B | 64.9 | 3.3 | 49.3 | 2.8 | 35.2 | 2.7 | 54.6 | 3.2 |
| BT-Adapter† (Liu et al., 2023c) | Vicuna-7B | 67.0 | 3.6 | 51.2 | 2.9 | 46.1 | 3.2 | – | – |
| LLaMA-VID (Li et al., 2023c) | Vicuna-13B | 70.0 | 3.7 | 58.9 | _3.3_ | 47.5 | **3.3** | – | – |
| Vista-LLaMA (Ma et al., 2023) | Vicuna-7B | 65.3 | 3.6 | _60.5_ | _3.3_ | 48.3 | **3.3** | 60.7 | _3.4_ |
| MovieChat (Song et al., 2023) | Llama 2-7B | 61.0 | 2.9 | 49.7 | 2.8 | **51.5** | 3.1 | 49.9 | 2.7 |
| Video Chat 2 (Li et al., 2024) | Vicuna-7B | 70.0 | **3.9** | 54.1 | _3.3_ | 49.1 | **3.3** | _61.7_ | – |
| Video-LaVIT (Jin et al., 2024) | Llama 2-7B | _73.2_ | **3.9** | 59.3 | _3.3_ | _50.1_ | **3.3** | – | – |
| Video-LLaVA (Lin et al., 2024) | Vicuna 1.5-7B | 70.7 | **3.9** | 59.2 | **3.5** | 45.3 | **3.3** | – | – |
| MiniGPT4-Video (Ataallah et al., 2024) | Llama 2-7B | 72.9 | 3.8 | 58.8 | _3.3_ | 45.9 | 3.2 | – | – |
| VaQuitA (Ours) | Llama 2-7B | **74.6** | 3.7 | **68.6** | _3.3_ | 48.8 | **3.3** | **62.3** | **3.5** |

Table 1: Zero-Shot question-answering performance comparison of `VaQuitA` with other models. Our `VaQuitA` demonstrates SOTA performance across all examined datasets.* denotes the results reported in Maaz et al. (2023) and † denotes the results reported in Liu et al. (2023c). The best performance in **bold** and the second best underlined.

relevant with the question. The multi-head design enables the exploration of the weight parameters in more feature subspaces for superior representation learning (Vaswani et al., 2017).

**VQ-Former Overview.** We use `Cross_Attn` to denote the Visual-Query Cross Attention, and the entire procedure of our VQ-Former can be written as:

$$O_a = \texttt{Cross\_Attn}(M, X), M' = O_a \cdot \tanh(g_{\text{attn}}) + MW_M, \tag{5}$$

$$O_f = \texttt{Feed\_Forward}(M'), M'' = O_f \cdot \tanh(g_{\text{ff}}) + M'. \tag{6}$$

Here the learnable parameter $W_M \in \mathbb{R}^{d \times d_{\text{text}}}$ is applied to transform the dimension of the video representative features into token feature dimension for the residual architecture. Here tanh denotes Hyperbolic Tangent function and `Feed_Forward` denotes a Feed Forward net block containing 2 linear layers with Layer Normalization and GELU (Hendrycks & Gimpel, 2016) activation layer. $g_{\text{attn}}$ is Attention Gate and $g_{\text{ff}}$ is FeedForward Gate, which are both learnable scalar parameters borrowed from Flamingo (Alayrac et al., 2022) for improved stability and performance. Eventually, the output question-interacted video features $M''$ are input to the LLM together with the input question embeddings. Different from the existing visual-text interleaving architectures, *e.g.*, Q-Former (Li et al., 2023a) or Gated Cross-Attention layer (Alayrac et al., 2022), our VQ-Former converts visual features to Queries and text features to Keys and Values for attention value computation. The underlying rationale of our approach is to utilize the information from the query as a directive to enhance the learning of pivotal visual embeddings. **This is significantly different from existing literature where visual features are converted to Keys and Values and text features are converted to Queries (Alayrac et al., 2022; Li et al., 2023a).** Also, the output of VQ-Former is concatenated to the question, differing from existing works where the output is directly sent to the language models.

## 3.3 End-to-end Training

Our `VaQuitA` supports end-to-end training: the trainable parameters include the Text Token Embedding Layer, the VQ-Former, and the Video Perceiver. The visual encoder (CLIP) and the Large Language model weights are derived from pretrained weights and are frozen during our training. The CLIP model employed to extract $f_{query}$ and $f_{video}$ is also frozen during the training. We employ the standard smoothed Negative Log-Likelihood Loss in NLP literature.

## 3.4 Prompt Engineering

Prompt engineering (Wei et al., 2022; Zhou et al., 2022; Gu et al., 2023) refers to the systematic design and modification of input prompts to guide machine learning models, particularly pretrained LLMs, to produce desired or more accurate outputs. The essence of this technique is rooted in the understanding that the input

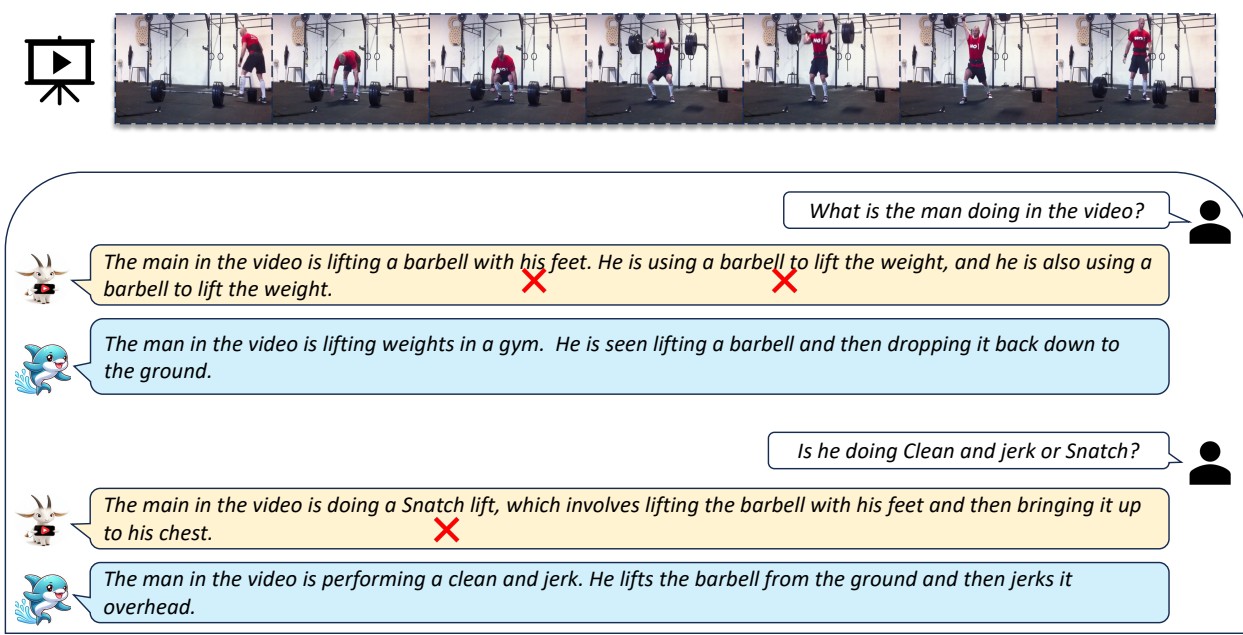

Figure 3: Given a video clip of a man lifting weights, we ask questions on what the man is doing and whether he is doing Clean and Jerk or Snatch. Our `VaQuitA` answers both the questions correctly. While the baseline Video-ChatGPT (Maaz et al., 2023) generates a repetitive answer to the first question, seeming somewhat chaotic, and fails to discriminate that the man is doing a Clean and Jerk, rather than Snatch.

provided to a model doesn't merely serve as a query but also as a form of soft guidance, potentially shaping the model's behavior and outputs. In our experiments, we are excited to discover that in the testing phase, if we add a prompt "Please be critical." before the question, zero-shot question answering performance can be significantly and consistently improved. This might imply an intriguing principle that, unlike in question answering in NLP the models are demanded to be calmer or more organized (Kojima et al., 2022; Yang et al., 2023b), the model needs to be more critical or judgmental for video question answering tasks. An ablation study on the prompts is carried out in Sec. 4.3.2 which verifies the implication.

## 4 Experiments

In the experimental implementation, we employ Llama 2 (7B) (Touvron et al., 2023b) as the foundational LLM backbone and initialize its weight using the weights of LLaVA-1.5 (Liu et al., 2023a). We fine-tune the trainable parameters in `VaQuitA` using the video instruction dataset VideoInstruct-100K[1] (Maaz et al., 2023), comprising roughly 100,000 pairs of video instructions. The fine-tuning phase spans three epochs, utilizing a step size of value $2e$-5 and a total batch size of value 32. For fair comparison, we keep the data-level hyperparameters as the same in literature: $T = 100, d = 1024, d_{text} = 4096,$. We employ the "clip-vit-large-patch14" CLIP version for video feature extraction. Specifically, for the sampling-period features $f_{query}$ and $f_{video}$, we use the last layer of the CLIP model with dimension 768. **The frame selection process is done before finetuning.** For the video feature extraction before the video perceiver, we utilize the last but one layer of CLIP with patch number $n = 256$ and feature dimension $d = 1024$. We chose $m = 356$ in Video Perceiver, which is the same as the dimension after spatio-temporal pooling in Video-ChatGPT (Maaz et al., 2023) for a fair comparison. The perceiver depth is set as $p = 1$. For all the attention blocks in both Video Perceiver and VQ-Former, we set $d_h = 64, H = 8$ and scale parameter $s_q = 8$. All the training experiments are conducted on eight A100 80GB GPUs. For testing, one GPU with 15 GB memory is sufficient.

---

[1] https://huggingface.co/datasets/MBZUAI/VideoInstruct-100K

| FA | DA | PE | MSVD | | MSRVTT | | Activity | |
|---|---|---|---|---|---|---|---|---|
| | | | A. | S. | A. | S. | A. | S. |
| ✗ | ✗ | ✗ | 65.1 | 3.3 | 49.9 | 2.8 | 42.5 | 3.0 |
| ✗ | ✗ | ✓ | 65.8 | 3.3 | 50.5 | 2.9 | 43.9 | 3.1 |
| ✗ | ✓ | ✗ | 64.5 | 3.2 | 50.8 | 2.9 | 44.9 | 3.1 |
| ✗ | ✓ | ✓ | 65.9 | 3.3 | 52.8 | 3.0 | 45.7 | 3.1 |
| ✓ | ✗ | ✗ | 70.8 | 3.5 | 59.7 | 3.1 | 47.4 | 3.1 |
| ✓ | ✗ | ✓ | 71.0 | 3.5 | 60.3 | 3.1 | 47.8 | 3.2 |
| ✓ | ✓ | ✗ | 74.4 | 3.7 | 68.5 | 3.3 | 47.7 | 3.3 |
| ✓ | ✓ | ✓ | **74.6** | **3.7** | **68.6** | **3.3** | **48.8** | **3.3** |

Table 2: Ablation of components. FA, DA, and PE signify Feature Alignment, Data Alignment, and Prompt Engineering. A. denotes accuracy and S. denotes score.

| t-Q-v-KV | v-Q-t-KV | DA | PE | MSVD | | MSRVTT | | Activity | |
|---|---|---|---|---|---|---|---|---|---|
| | | | | A. | S. | A. | S. | A. | S. |
| ✓ | | ✗ | ✗ | 66.8 | 3.3 | 52.7 | 2.9 | 44.3 | 2.9 |
| | ✓ | ✗ | ✗ | 70.8 (+4.0) | 3.5 (+0.2) | 59.7 (+7.0) | 3.1 (+0.2) | 47.4 (+3.1) | 3.1 (+0.2) |
| ✓ | | ✗ | ✓ | 67.4 | 3.3 | 54.1 | 3.0 | 45.2 | 3.0 |
| | ✓ | ✗ | ✓ | 71.0 (+3.6) | 3.5 (+0.2) | 60.3 (+6.2) | 3.1 (+0.1) | 47.8 (+2.6) | 3.2 (+0.2) |
| ✓ | | ✓ | ✗ | 67.9 | 3.4 | 56.2 | 3.0 | 45.3 | 3.0 |
| | ✓ | ✓ | ✗ | 74.4 (+6.5) | 3.7 (+0.3) | 68.5 (+12.3) | 3.3 (+0.3) | 47.7 (+2.4) | 3.3 (+0.3) |
| ✓ | | ✓ | ✓ | 68.2 | 3.4 | 56.5 | 3.0 | 45.8 | 3.1 |
| | ✓ | ✓ | ✓ | **74.6** (+6.4) | **3.7** (+0.3) | **68.6** (+12.1) | **3.3** (+0.3) | **48.8** (+3.0) | **3.3** (+0.2) |

Table 3: Comparison of our cross-attention computation and the traditional approach. Here t-Q-v-KV denotes that text features serve as queries and video features serve as keys and values; v-Q-t-KV denotes that video features serve as queries and text features serve as keys and values. A. denotes accuracy and S. denotes score.

## 4.1 Zero-shot Video Question Answering

We carry out an exhaustive quantitative assessment using several prevalent open-ended video question-answer datasets, encompassing MSRVTT-QA (Xu et al., 2017; Lei et al., 2023), MSVD-QA (Xu et al., 2017; Lei et al., 2023), Activity Net-QA (Yu et al., 2019), NeXT-QA Xiao et al. (2021). Following Maaz et al. (2023), the assessments are performed in a zero-shot setting, utilizing GPT-guided evaluation to gauge the model's proficiency. This assessment method calculates the precision of the model's predicted outputs (accuracy) and ranks them on a 1-5 scale (score). To ensure a fair comparison with the baselines, we employ Azure GPT-3.5-turbo API for evaluation, which is consistent with Maaz et al. (2023). Our `VaQuitA`'s efficacy is juxtaposed with other notable models, namely FrozenBiLM (Yang et al., 2022), VideoLLaMA (Zhang et al., 2023), LLaMA-Adapter (Gao et al., 2023), Video Chat (Li et al., 2023b), Video-ChatGPT (Maaz et al., 2023), BT-Adapter (Liu et al., 2023c), LLaMA-VID (Li et al., 2023c), Vista-LLaMA (Ma et al., 2023), MovieChat Song et al. (2023), Video Chat 2 (Li et al., 2024), Video-LaVIT (Jin et al., 2024), Video-LLaVA (Lin et al., 2024) and MiniGPT4-Video Ataallah et al. (2024). From Tab. 1, we can draw the conclusion that `VaQuitA` achieves SOTA performance in both accuracy and score across the three benchmark datasets.

## 4.2 Multi-Round Conversation

The experiments conducted predominantly address scenarios involving a singular question and answer. However, in practical applications such as Copilot or assistants for industrial products, the capacity for multi-round conversations is crucial for user experience. To evaluate this aspect, we compare the multi-round conversation capabilities of `VaQuitA` with one of the baselines Video-ChatGPT (Maaz et al., 2023). As depicted in Fig. 3, `VaQuitA` demonstrates consistently more accurate and comprehensive conversational abilities compared to Video-ChatGPT. This highlights `VaQuitA`'s potential for industrial applications. More video dialogue examples are provided in supplementary.

## 4.3 Ablation Studies

### 4.3.1 Ablation of the components of `VaQuitA`

We perform ablation studies w.r.t. the components of `VaQuitA` including the Data Alignment, Feature Alignment, and Prompt Engineering. We conduct experiments on all three Video Question Answering datasets. We also list the baseline results using Llama 2 as the LLM backbone without all three modules, in which one MLP layer is used to project the visual embeddings to the token space as Maaz et al. (2023). Tab. 2 shows that Data Alignment, Feature Alignment, and Prompt Engineering all contribute to zero-shot video QA performance. On one hand, the performance of merely adopting Feature Alignment performs better than merely adopting Data Alignment without Prompt Engineering, implying that feature-level learning is comparatively more significant than input data selection for our task. On the other hand, with Prompt Engineering, the model will degrade a lot without Data Alignment. In addition, Prompt Engineering improves the performance in all cases. We also conduct experiments ablating the Video Perceiver (VP) and

| Length | 0 | 20 | 40 | 60 | 80 | 100 |
|---|---|---|---|---|---|---|
| **Acc.** (↑) | 47.8 | 48.2 | 48.5 | **49.0** | 48.1 | 47.5 |
| **Score** (↑) | 3.2 | 3.3 | 3.3 | **3.3** | 3.3 | 3.2 |

Table 4: Ablation of similarity-based sampling frame #.

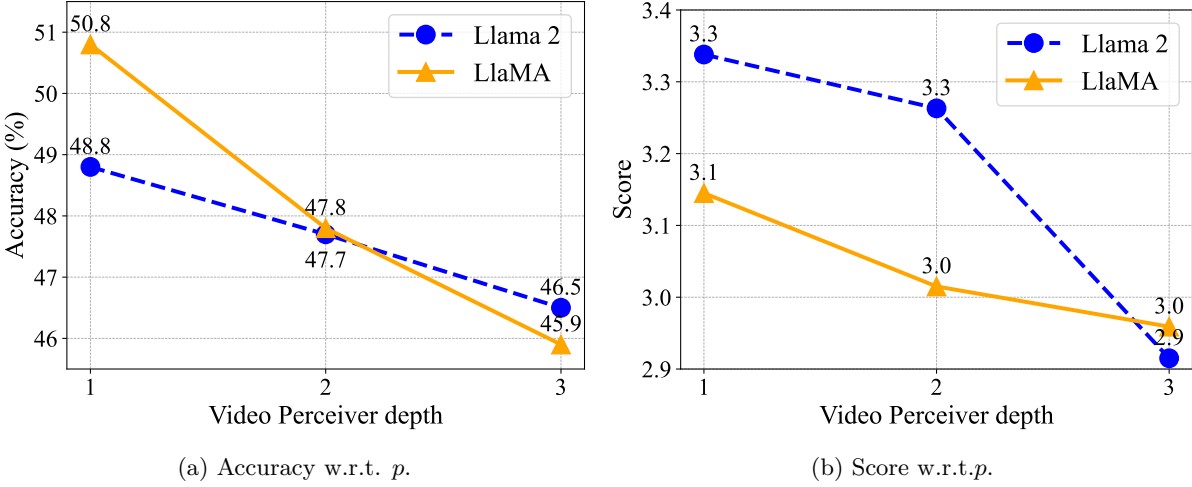

(a) Accuracy w.r.t. $p$.

(b) Score w.r.t. $p$.

Figure 4: Performance on Activity Net-QA (Yu et al., 2019) using pretrained Llama 2 (Touvron et al., 2023b) and LLaMA (Touvron et al., 2023a).

VQ-Former respectively. When ablating VP, we directly pool the CLIP features along spatial dimension and temporal dimension and then concatenate the features as Maaz et al. (2023) does. As is shown in Tab. 5 in Appendix B, both VP and VQ-Former contribute much to the performance and the combination of them leads to the best results. We further compare our strategy of converting video features to Quries and prompt features to Keys and Values with the strategy of converting prompt features to Queries and video features to Keys and Values. The architecture is kept the same and the results in Tab. 3 indicate that our approach manifests obvious superiority whether DA or PE is used consistently. **This largely arises from the fact that because the output of the cross-attention layer is concatenated with the prompt tokens to be sent to LLM, video is the primary context for which we want to enhance or refine representations using information from textual modality.** Therefore video features should serve as Queries while text features serve as Keys and Values.

### 4.3.2 Ablation of Hyperparameters

We further study the effects of changing the hyperparameter values in our `VaQuitA` framework. We conduct the ablation studies on the Activity Net-QA testing dataset.

**Similarity-based Sampling Frame Number.** We employed a mixed strategy of sampling to focus on question-related frames while looking broadly for performance stability. CLIP features are not perfect and it is likely that the CLIP-similarity selected frames are not the places of interest. We provide additional experiments by changing the similarity-based sampling frame number. As shown in Tab. 4, sampling completely uniformly or completely based on CLIP feature similarities gives the inferior performance. **This means that the model should both look broadly and focus on certain frames of interest across the temporal dimension to reach the best performance in the data input phase.**

**Video Perceiver Depth & Pretrained Model.** We try using multiple layers in Video Perceivers and using the LLaMA (Touvron et al., 2023a) model with weight initialization from LLaVA (Liu et al., 2023b). As illustrated in Fig. 4, the accuracy of `VaQuitA` drops when the layer number $p$ of the Video Perceiver increases for both LLaMA and Llama 2 backbone. This might largely result from the small training epoch we use and the limited size of training data. For the LLM weights initialized from LLaVA and LLaVA-1.5, we find that the performance gap is not as large as expected, and using LLaMA (LLaVA-1.5) pretrained weights with one layer in Video Perceiver even achieve 50.8 accuracy on Activity Net-QA dataset, **much better**

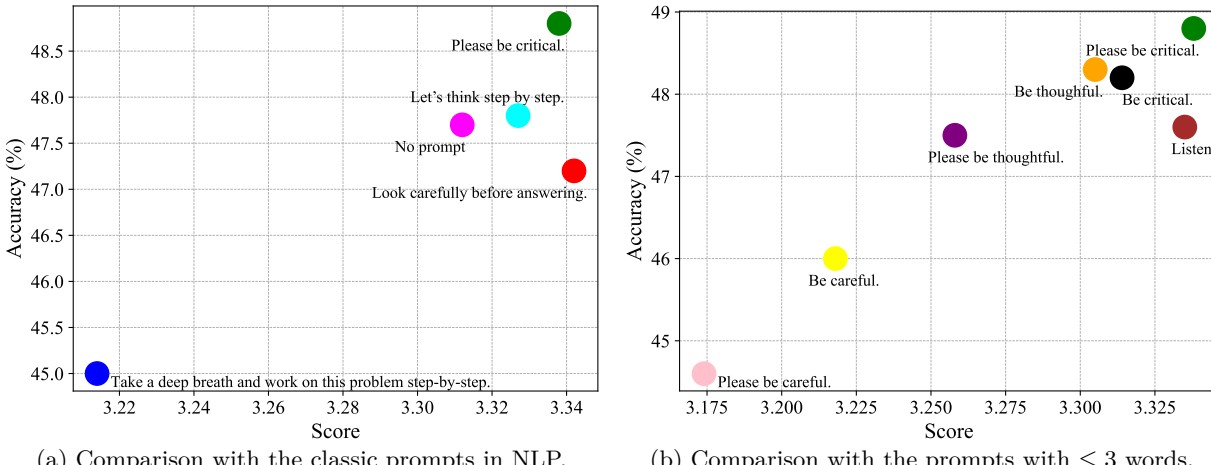

(a) Comparison with the classic prompts in NLP.    (b) Comparison with the prompts with ≤ 3 words.

Figure 5: Accuracy and score results on Activity Net-QA (Yu et al., 2019) dataset of different prompt designs.

**than the baseline Video-ChatGPT Maaz et al. (2023)**. On the other hand, models initialized using LLama 2 are more robust to the perceiver depth and are significantly better in relative score evaluation.

**Prompt Engineering Design.** We ablate the prompt added before the question. We compare our designed prompt with two popular instruction prompts in NLP: "Take a deep breath and work on this problem step-by-step." (Yang et al., 2023b) and "Let's think step by step." (Kojima et al., 2022). We also compare with another prompt "Look carefully before answering." and indicate the performance when not adding a prompt. From the accuracy and score results shown in Fig. 5a, we can draw the conclusion that our designed prompt, "Please be critical.", performs the best with both the highest accuracy and the highest score. "Let's think step by step." (Kojima et al., 2022) improves the performance slightly while "Take a deep breath and work on this problem step-by-step." (Yang et al., 2023b) degrades the performance. We conduct additional experiments with different prompts that have fewer than or equal to 3 words: "Please be careful." (pbc), "Please be thoughtful." (pbt), "Be critical." (bcr), "Be thoughtful." (bt), "Be careful." (bca), "Listen." (l). As shown in Fig. 5b, our prompt "Please be critical." exhibits the best performance. Also, "Be critical." also exhibits nearly excellent performance, implying that the word "critical" is significant.

## 5  Conclusion

Our proposed `VaQuitA` represents a significant stride in video understanding. By moving away from traditional frame sampling methods and adopting a CLIP-score guided technique, we have achieved a more nuanced and effective integration of video frame and text data. The innovative combination of a trainable video perceiver with a visual-query transformer mechanism allows for a dynamic interplay between video features and input questions, further augmented by the strategic use of prompts. The results shown in the paper clearly demonstrate that `VaQuitA` not only excels in zero-shot video question-answering tasks but also in generating coherent and contextually rich multi-turn video dialogues.

## Broader Impact Statement

We confirm that this paper strictly follows ethical research standards. All of the references mentioned in the main paper are publicly available, and no human participants or personally identifiable information have been included. The goal of the paper is to introduce new alignment techniques in LLM-based methods for enhancing video understanding performance. All previous related works have been cited appropriately, with due recognition given to their original contributions.

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

# Critical Video-Language Understanding via Query-Guided Frame Selection and Visual-Query Transformation

## Supplementary Material

## A   Data Alignment Figure

The illustration of our proposed frame sampling approach is shown in Fig. 6.

## B   Ablation Study of Video Perceiver and VQ-Former in Feature Alignment.

The performance when ablating the Video Perceiver or the VQ-Fromer in the Feature Alignment module is presented in Tab. 5.

## C   Raw Videos of Sec. 4.2

We supplement the raw videos of the two examples in Sec. 4.2 of the main paper, namely "multi_round_example_1.mp4" and "multi_round_example_2.mp4". They are chosen from the test set of ActivityNet-200 (Caba Heilbron et al., 2015) dataset.

## D   `VaQuitA` Assistant Demo

We also provide a video demo recording of our `VaQuitA` Assistant on Gradio (Abid et al., 2019) ("VaQuitA_demo.mp4"). The three example videos are chosen from the test set of TGIF (Li et al., 2016), Social-IQ 2.0 and ActivityNet-200 (Caba Heilbron et al., 2015) datasets. The videos are about a boy falling down the skateboard on a ramp, a doctor and patient talking to each other in the hospital and a man shaving himself in the bathroom, respectively. We show in our demo recording that the `VaQuitA` Assistant is able to generate high-quality multi-round conversations at a high responding speed. It is able to precisely summarize the content of a video, identify the relationships between characters and events, and pinpoint locations.

## E   Test-time Data Alignment

We conduct an additional experiment using our proposed sampling approach in Sec. 3.1 during the inference stage. We use the Video-ChatGPT (Maaz et al., 2023) trained model and only change the sampling way in inference. The baseline is uniform sampling. Given a video clip of MSNBC news report (video given at "test_time_da_example.mp4"), we ask a video question: "During the movie, there is a video clip with flying animals. What is the flying animal, bird or bat?" for 3 independent times. The correct answer is "bat", which corresponds to 2:16-2:22 time stamp of the video. For uniform sampling, the model answers: "The flying animal in the video is a bird." for 3 times, which is wrong; for our proposed sampling method, the model answers: "The flying animal in the video is a bat" for 3 times, which is correct.

This superiority of our Data Alignment module mainly results from the CLIP Feature Similarity-based Frame Selection component, which is verified by checking the selected frames. We supplement the directories of the sampled frame of uniform sampling and our data alignment sampling method. The sampled frames using uniform sampling are stored under directory "uniform_sampled_frames" and the sampled frames using our proposed sampling method are under directory "ours_sampled_frames". We can see that the uniform sampling only samples one frame ("frame_4223.jpg") related to the question, while our proposed sampling method samples 13 related frames ("frame4197.jpg", "frame4198.jpg", "frame4201.jpg", "frame4206.jpg", "frame4207.jpg", "frame4247.jpg",
"frame4280.jpg", "frame4281.jpg", "frame4282.jpg", "frame4287.jpg",

| Dataset | VP | VQ-Former | Acc. (↑) | Score (↑) |
|---------|----|-----------|----------|-----------|
| MSVD | ✗ | ✗ | 65.3 | 3.2 |
|  | ✓ | ✗ | 68.5 | 3.4 |
|  | ✗ | ✓ | 70.9 | 3.5 |
|  | ✓ | ✓ | **74.6** | **3.7** |
| MSRVTT | ✗ | ✗ | 51.0 | 2.9 |
|  | ✓ | ✗ | 59.4 | 3.1 |
|  | ✗ | ✓ | 62.3 | 3.1 |
|  | ✓ | ✓ | **68.6** | **3.3** |
| Activity | ✗ | ✗ | 45.7 | 3.1 |
|  | ✓ | ✗ | 46.9 | 3.2 |
|  | ✗ | ✓ | 47.4 | 3.2 |
|  | ✓ | ✓ | **48.8** | **3.3** |

Table 5: Ablation of Video Perceiver and VQ-Former in Feature Alignment.

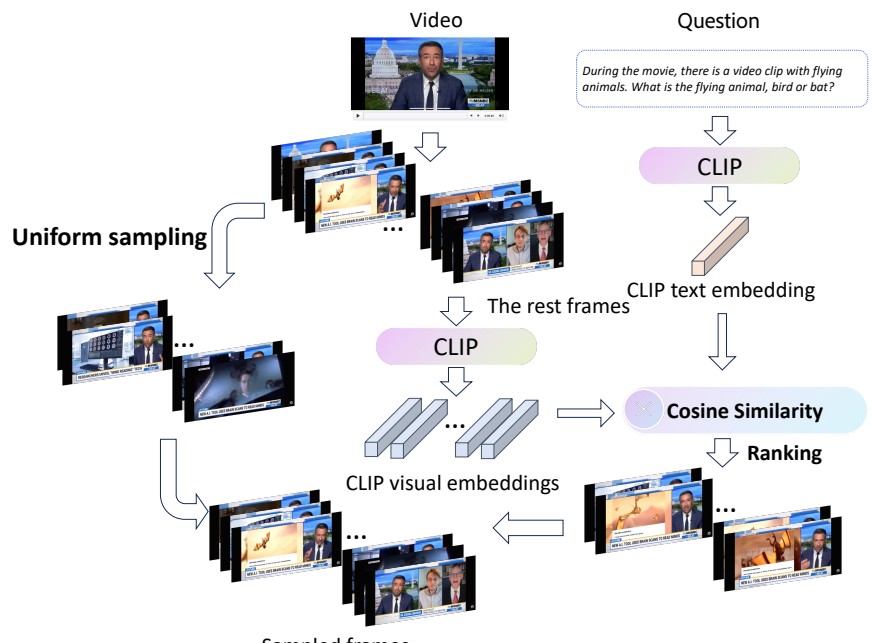

Figure 6: **Data alignment.** Our proposed sampling module consists of both uniform sampling and similarity-based sampling for the training process.

"frame4197.jpg", "frame4288.jpg", "frame4289.jpg", "frame4304.jpg"). Since our sampling method samples more frames corresponding to the question, the model can answer more correctly, which reflects the effect of Data Alignment in the inference phase.

## F    More Multi-round Conversation Examples Compared with Video-ChatGPT (Maaz et al., 2023)

From Fig. 7 to Fig. 13, we supplement more multi-round conversation examples in addition to Sec. 4.2 with their raw videos, namely
"multi_round_example_3.mp4",
"multi_round_example_4.mp4",
"multi_round_example_5.mp4",
"multi_round_example_6.mp4",
"multi_round_example_7.mp4", and "multi_round_example_8.mp4". They are chosen from the test set of ActivityNet-200 (Caba Heilbron et al., 2015) dataset. It is shown that `VaQuitA` generates consistently better

answers and exhibits significantly stronger video understanding ability compared to Video-ChatGPT (Maaz et al., 2023).

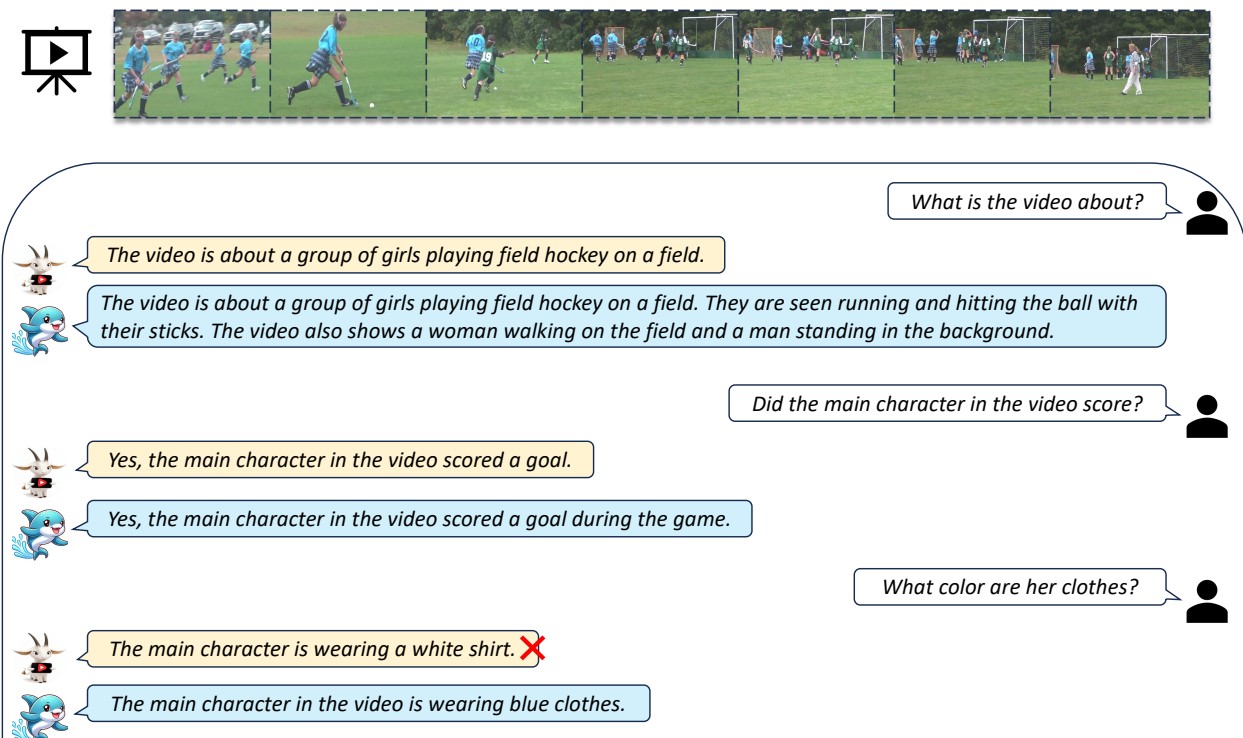

Figure 7: Given a video clip on a group of girls playing field hockey, we ask questions on the content of the video, whether the main character scores, and the color of the main character's clothes. Our `VaQuitA` can answer all the questions correctly while the baseline Video-ChatGPT (Maaz et al., 2023) fails to tell the correct color of the clothes the girl is wearing (marked by red cross). In addition, the generated answers of `VaQuitA` are more detailed and specific like a human chatting with the user, while the responses of Video-ChatGPT are short and like being forced to complete a task.

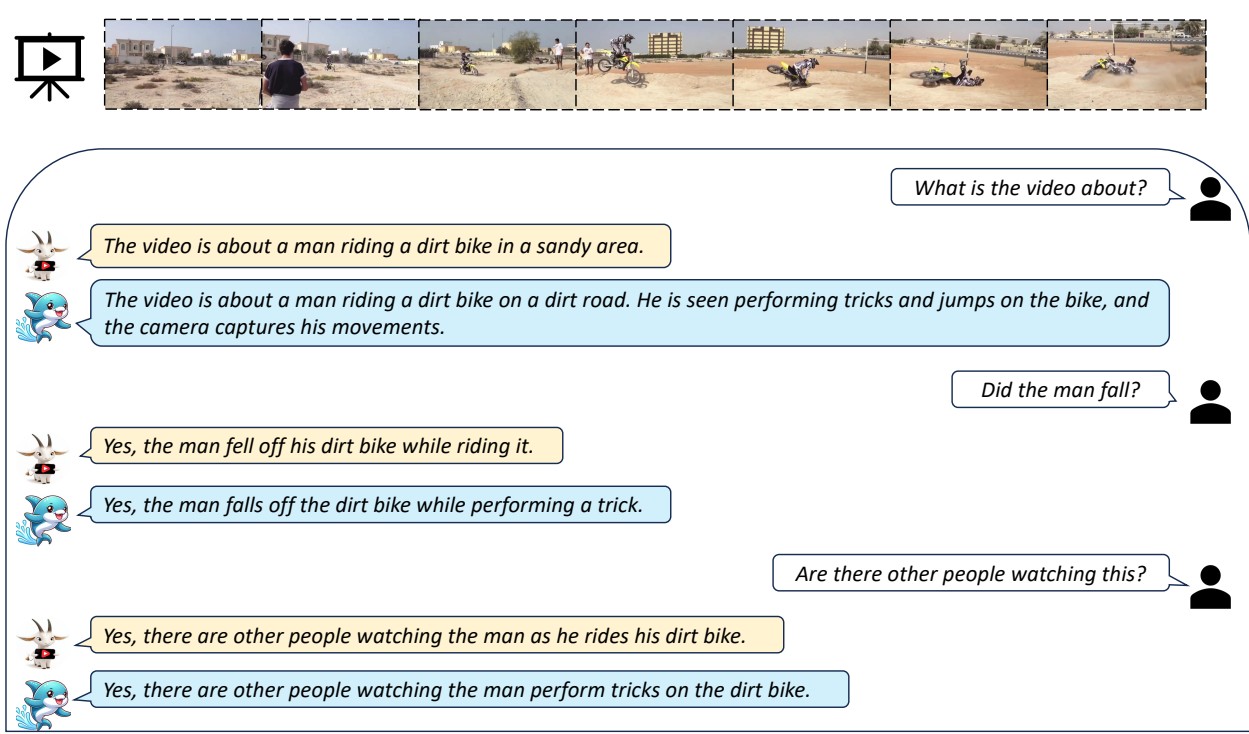

Figure 8: Given a video clip on a man riding a dirt bike, we ask questions on the content of the video, whether the man fell, and whether there are other people watching this. Our `VaQuitA` can answer all the questions correctly and can identify that the person was trying to do a trick when riding, which leads to his falling. In contrast, the baseline Video-ChatGPT (Maaz et al., 2023) fails to tell that the man fell when doing a trick when riding. In addition, the generated answers of `VaQuitA` are more detailed and specific.

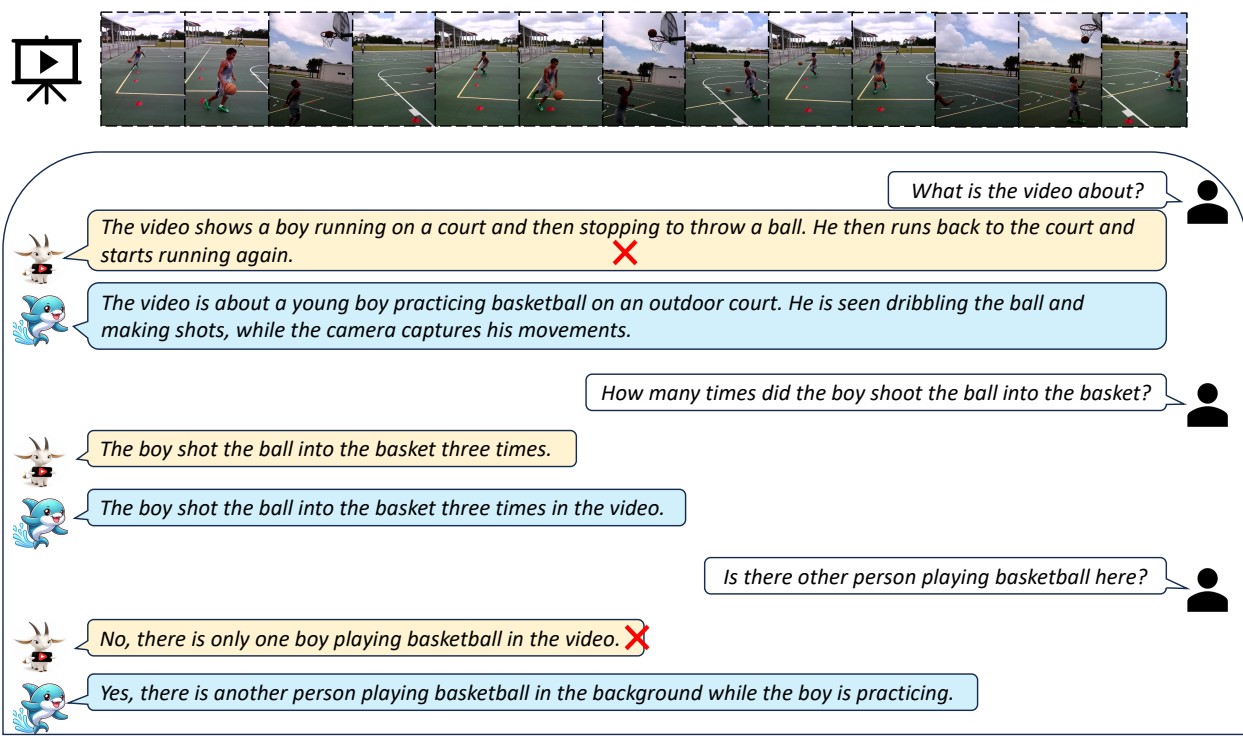

Figure 9: Given a video clip on a boy playing basketball on an outside court, we ask questions on the content of the video, how many times the boy shot the ball, and whether there is another person playing basketball there. Our `VaQuitA` can identify that the boy was practicing basketball with dribbling and shooting and tell that there is another man playing basketball behind the boy. In contrast, the baseline Video-ChatGPT (Maaz et al., 2023) fails to identify the movements of the boy well and gives the wrong answer on the last question. In addition, the generated answers of `VaQuitA` are more detailed and specific.

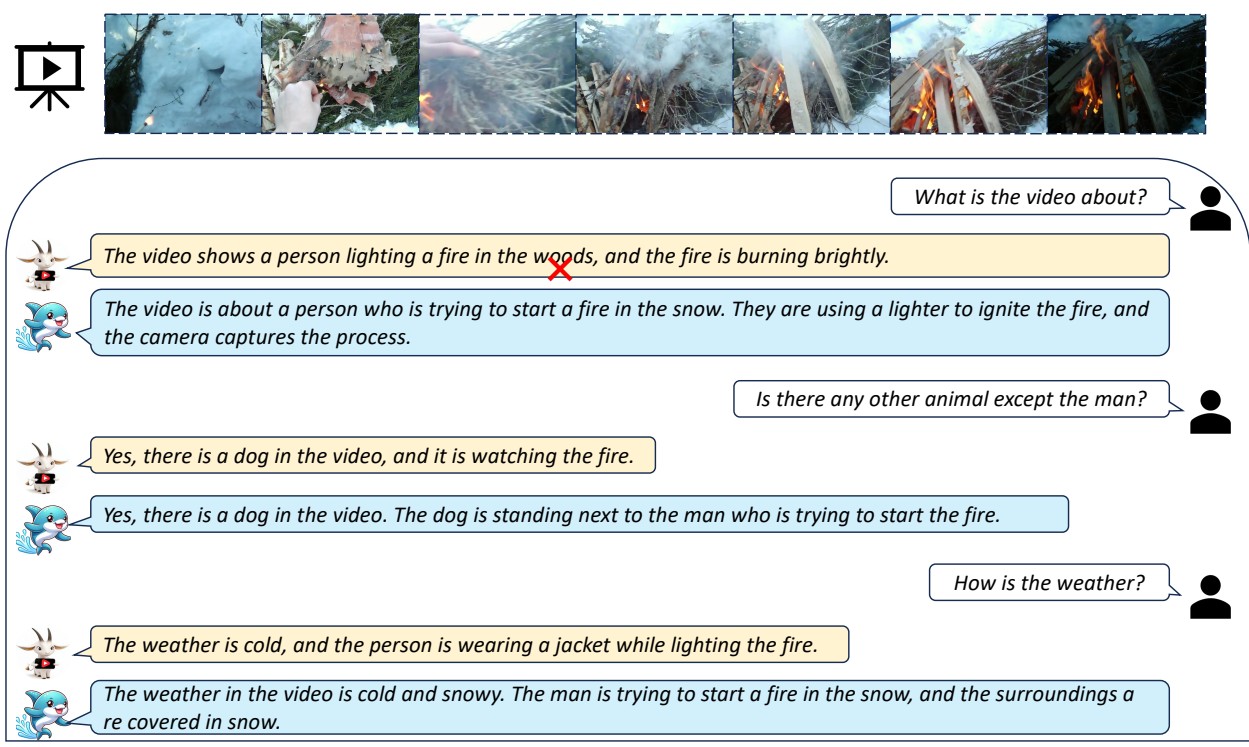

Figure 10: Given a video clip of a person making a fire in the snow, we ask questions on the content of the video, whether there is another animal, and how the weather is. Our `VaQuitA` can answer all the questions correctly and can identify that it is snowy and the fire is started in the snow. In contrast, the baseline Video-ChatGPT (Maaz et al., 2023) fails to tell the snowy weather and mistakenly identifies that the fire is lit in the woods. In addition, the generated answers of `VaQuitA` are more detailed and specific.

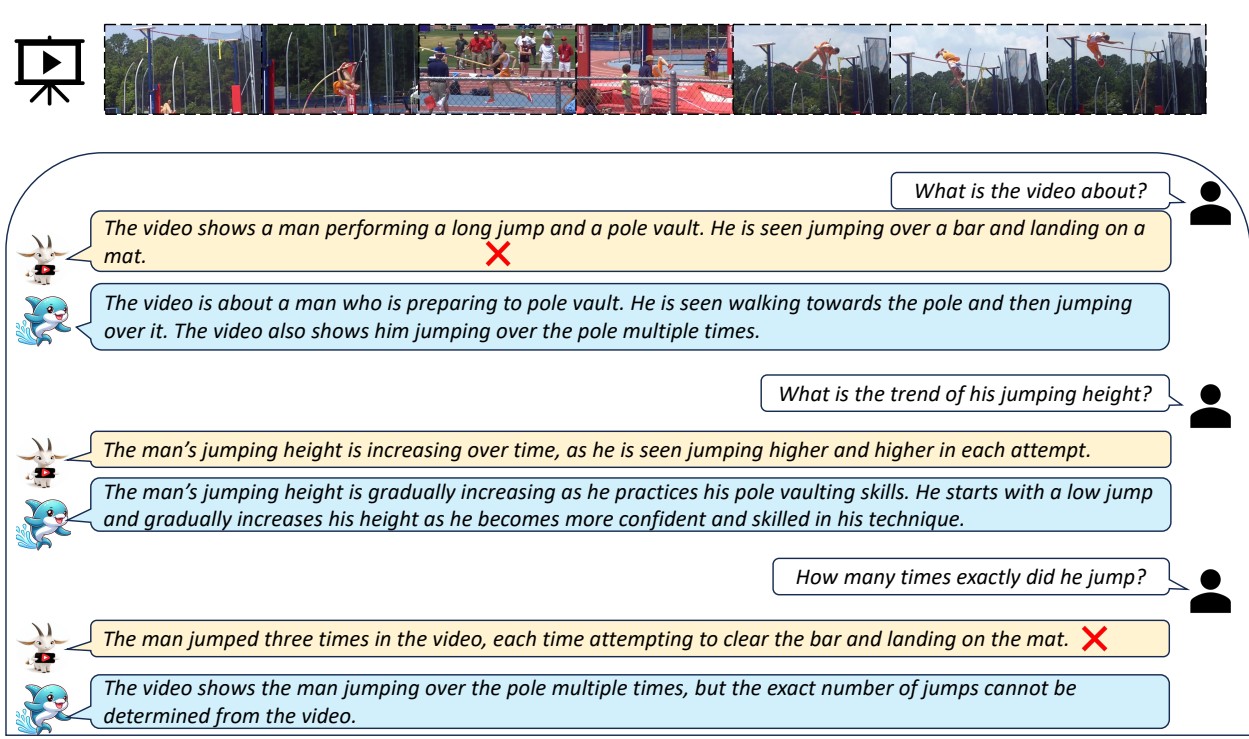

Figure 11: Given a video clip of a man doing pole vault, we ask questions on the content of the video, the trend of his jumping height, and the times of his jumping. Our `VaQuitA` can answer most of the questions correctly, except the third one as the man jumped four times in total. The baseline Video-ChatGPT (Maaz et al., 2023) gives the wrong answer to the third question, either. The advantage of `VaQuitA` is that for the first question on the content of the video, it can identify that the man jumped over the pole multiple times, which is impressive. Note that it is bearable to fail to remember how many times in total the man jumped, which is also challenging even for a human being. In addition, the generated answers of `VaQuitA` are more detailed and specific.

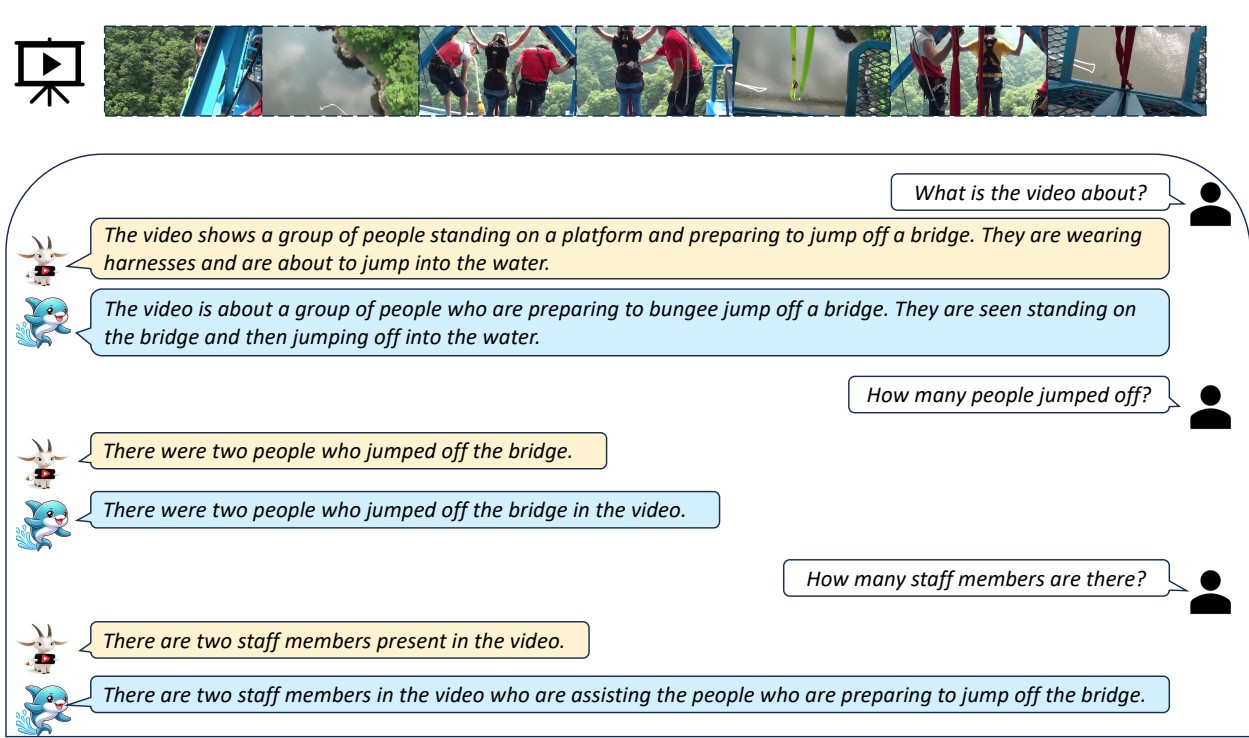

Figure 12: Given a video clip of a group of people doing bungee jump off a bridge, we ask questions on the content of the video, the number of people jumped, and the number of staff members. Our VaQuitA can answer all the questions correctly and can identify that the sports is bungee jump. In contrast, the baseline Video-ChatGPT (Maaz et al., 2023) fails to tell the name of the sports the people are doing. In addition, the generated answers of VaQuitA are more detailed and specific.

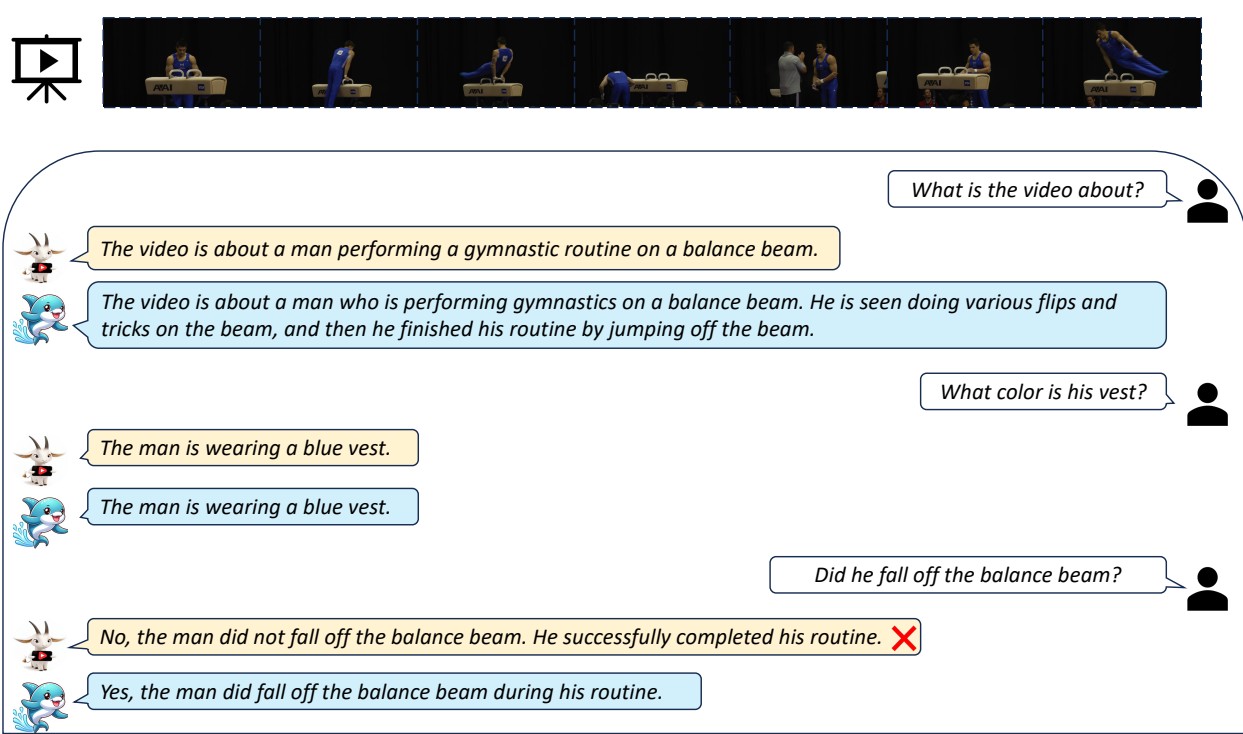

Figure 13: Given a video clip of a man performing gymnastics on a balance beam, we ask questions on the content of the video, the color of his vest, and whether he fell off the balance beam. Our VaQuitA can answer all the questions correctly. In contrast, the baseline Video-ChatGPT (Maaz et al., 2023) fails to find that the man actually fell off the balance beam once. In addition, the generated answers of VaQuitA are more detailed and specific in describing the movements of the gymnastic, especially for the first question.

