# OpenReview forum: "Critical Video-Language Understanding via Query-Guided Frame Selection and Visual-Query Transformation"
_TMLR — Withdrawn by Authors_

### Review · Reviewer_YD45 · 2026-03-22

**Summary Of Contributions:**

The paper proposes VaQuitA, a video large language model that improves alignment between video features and text. It introduces CLIP score-based frame selection for query-relevant sampling, a video perceiver with a visual-query transformer for feature alignment, and a simple prompting strategy ("Please be critical."). The method is evaluated on zero-shot video QA and multi-turn dialogue.

Strengths
- Addresses an important problem of video-text alignment.
- Query-aware frame selection is a reasonable and intuitive idea.

Weaknesses
- Uses an outdated design (video perceiver) compared to recent simpler and more effective approaches.
- Data alignment strategy may harm temporal modeling without explicit timestamp encoding; CLIP text encoder is limited for long context reasoning.
- It is unclear why mixed sampling (uniform sampling + clip-based sampling) during training improves performance when evaluation uses uniform sampling only; this requires further clarification and broader validation.
- Prompting gain is marginal.
- Multi-turn dialogue claims lack quantitative validation.
- Evaluation benchmarks are limited.
- Overall, the paper feels outdated, with old baselines and backbones and missing comparisons to recent models.

**Audience:**

No

**Audience Explanation:**

Critically, this paper feels quite outdated. The baselines used for comparison are all models from 2-3 years ago, and both the LLM backbone (Llama2) and the vision encoder (CLIP) used in VaQuitA have since been surpassed by more recent, higher-performing models (e.g., Qwen3, SigLIP 2). Therefore, contrary to the paper's claim of achieving state-of-the-art performance, the method underperforms significantly compared to recent models such as PLM, InternVL 3.5, GLM-4.5V, MiniCPM-V-4.5, Eagle2.5 and Molmo2. Updating the backbone models and conducting comparison with these more recent approaches would be necessary to attract attention from the research community.

**Broader Impact Concerns:**

The paper has addressed the broader impact concerns of the work.

**Claims And Evidence:**

No

**Claims Explanation:**

1. One of the main contributions of this paper is the feature alignment module, which adopts a video perceiver. Until about 3-4 years ago, attentive resamplers such as the perceiver were widely used to encode video embeddings. However, (i) important inductive biases of video -- such as 2D spatial structure and temporal ordering -- can be lost during the abstraction process; (ii) the context length of backbone LLMs has increased dramatically; and (iii) the use of of perceivers often leads to training instability and slower convergence. As a result, recent models such as Qwen3-VL [1] tend to adopt adaptive pooling followed by an MLP projection. In this setting, unlike approaches such as VideoChatGPT that pool over the spatial or temporal dimensions, the pooling stride is smaller than the full spatial or temporal extent. In fact, [2] shows that simple 2D average pooling outperforms attentive resampling.
2. There are also two major issues with the data alignment module. First, uniform sampling or the clip-based frame selection proposed in the paper use different sampling rates for each video. As a result, relying solely on fixed time encoding -- as in the current model -- may make it difficult for the model to capture temporal dynamics. In fact, [3] points out the sub-optimal performance of such sampling strategies. Therefore, if clip-based frame selection is to be used, it should be complemented by encoding the timestamps of the sampled frames, as done in recent models such as Qwen3-VL and Molmo2 [4]. Second, the text encoder used in CLIP has a limited context length and primarily captures high-level semantics. Consequently, it may not be well-suited for tasks such as multi-turn dialogue, where the text context becomes long, or for questions that require complex multi-hop reasoning.
3. It is not immediately clear why performance improves on the evaluation benchmarks when mixed sampling, including clip-based sampling, is used only during training, while uniform sampling is used during evaluation. This requires further clarification, and in particular, more thorough validation across a broader range of benchmarks would be necessary.
4. The performance gain from the proposed prompt, "Please be critical.", is marginal across all evaluated benchmarks.
5. To demonstrate superiority in multi-turn dialogue, it is necessary to provide quantiative results in addition to a few qualitative examples, as presented in the paper. In particular, a human preference evaluation against baselines should have been conducted on a validation set consisting of at least several hundred examples.
6. The current evaluation benchmarks are insufficient to fully assess the performance of video LLMs. To better capture a broader range of task types and model capabilities, it is necessary to expand the evaluation suite: for example, by including benchmarks such as PerceptionTest, MVBench, Video-MME, and MLVU.

[1] Bai et al., Qwen3-VL Technical Report, arXiv preprint arXiv:2511.21631, 2025.

[2] Chung et al., Unifying Specialized Visual Encoders for Video Language Models, ICML 2025, 2025.

[3] Zohar et al, Apollo: An Exploration of Video Understanding in Large Multimodal Models, arXiv preprint arXiv:2412.10360, 2024.

[4] Clark et al., Molmo2: Open Weights and Data for Vision-Language Models with Video Understanding and Grounding, arXiv preprint arXiv:2601.10611, 2026.

**Requested Changes:**

Please incorporate responses to the above weaknesses into the draft. In particular, comparisons with recent models and evaluation on a broader range of benchmarks are essential.

---

### Review · Reviewer_AzBo · 2026-03-26

**Summary Of Contributions:**

This paper investigates improving the alignment between video representations and text queries in Vision-Language Models (VLMs), proposing a VaQuitA architecture. Specifically, it treats video features as queries and text features as keys and values, utilizing a Video Perceiver to capture video characteristics. Additionally, CLIP is employed for data filtering. Experiments are conducted on MSVD-QA, MSRVTT-QA, ActivityNet-QA, and NExT-QA using Llama-2 as backbone.

Strengths:
1. This paper addresses an important problem with significant practical value.
2. The proposed idea is intuitive and logically sound.

Weaknesses:
1. Some benchmark results reported in this paper still fall short of state-of-the-art (SOTA) performance. Please discuss the potential reasons for this gap.
2. The LLM backbone used appears relatively dated. Do you have any experimental results on more recent models, such as Qwen3 or Qwen3.5?

**Audience:**

Yes

**Audience Explanation:**

The paper demonstrates moderate novelty and offers some value to the research community."

**Claims And Evidence:**

Yes

**Claims Explanation:**

The paper features comprehensive experiments conducted on MSVD-QA, MSRVTT-QA, ActivityNet-QA, and NExT-QA using Llama-2 as the backbone.

**Requested Changes:**

1. Add results using different backbone models.
2. Discuss the reasons for the performance gap relative to state-of-the-art results.

---

### Review · Reviewer_WBFb · 2026-04-29

**Summary Of Contributions:**

This manuscript proposes VaQuitA, a video-language understanding framework that improves the interaction between video content and textual queries through three main components: (1) Query-guided frame selection (for data alignment), a hybrid sampling strategy combining uniform sampling with CLIP-based similarity to select frames relevant to the query. (2) Feature alignment via VQ-Former, which is a cross-attention module where video features act as queries and text features as keys/values, coupled with a Video Perceiver for feature compression. (3) Test-time prompt engineering, which is a simple prompt (“Please be critical.”) that improves zero-shot video QA performance.

Key Strengths:

- Clear and practical motivation (improving query-video alignment). Coherent integration of data-level and feature-level alignment.
- Consistent empirical improvements across benchmarks. Reasonably thorough ablation studies.

Key Weaknesses:

- Limited conceptual novelty; most components are adaptations of existing methods.
- Heavy reliance on heuristic prompt engineering.
- Some design choices lack strong theoretical justification.
- Limited analysis of failure cases and generalization.

**Audience:**

Yes

**Audience Explanation:**

Researchers working on video-language models, multimodal alignment, and vision-language reasoning, would likely find this work relevant.

**Broader Impact Concerns:**

No major ethical concerns are apparent.

The work focuses on improving video-language understanding and does not introduce new datasets involving sensitive data. However, potential downstream risks include:
- Misinterpretation of video content in real-world applications,
- Over-reliance on LLM-based reasoning without robustness guarantees.

The existing Broader Impact discussion is generally adequate, though it could briefly mention risks related to hallucination or incorrect reasoning in video understanding systems.

**Claims And Evidence:**

Yes

**Claims Explanation:**

Partially yes, but not fully convincing.

The manuscript currently provides:

- Quantitative comparisons on standard benchmarks showing improved performance.
- Ablation studies demonstrating the contribution of individual components.
- Qualitative examples for multi-turn dialogue.

However, several limitations weaken the strength of evidence:

- Comparisons rely partly on reported results rather than fully controlled re-implementations.
- Evaluation depends on GPT-based scoring, which may introduce bias and variance.
- Limited statistical analysis or robustness checks (e.g., across prompts or datasets).
- Insufficient investigation of failure cases and edge scenarios.

Overall, while the evidence supports performance gains, it is not fully sufficient to justify strong claims of superiority or generality.

**Requested Changes:**

RC1: Clarify and moderate novelty claims. Better position contributions relative to prior work (e.g., retrieval-based sampling, cross-attention architectures).

RC2: Strengthen experimental rigor. Include more controlled comparisons (same backbone, same data). Also, please provide statistical significance or variance analysis.

RC3: Expand failure case analysis. Analyze when query-guided sampling or VQ-Former fails, especially under ambiguous or noisy queries.

RC4: Justify key design choices. Please provide clearer reasoning or ablations for: (1) Attention direction (video as queries). (2) Hybrid sampling strategy. (3) Prompt selection.

---
Non-critical (would strengthen the work):

RC5: Evaluate generalization on more diverse datasets or longer videos.

RC6: Provide efficiency analysis (runtime, memory).

RC7: Explore robustness to different prompts or adversarial queries.

RC8: Improve clarity of writing and figures.

---

### Note · Authors · 2026-05-25

I have read and agree with the venue's withdrawal policy on behalf of myself and my co-authors.